# Exploration of Neusilin^®^ US2 as an Acceptable Filler in HPMC Matrix Systems—Comparison of Pharmacopoeial and Dynamic Biorelevant Dissolution Study

**DOI:** 10.3390/pharmaceutics14010127

**Published:** 2022-01-05

**Authors:** Tomáš Bílik, Jakub Vysloužil, Martina Naiserová, Jan Muselík, Miroslava Pavelková, Josef Mašek, Drahomíra Čopová, Martin Čulen, Kateřina Kubová

**Affiliations:** 1Department of Pharmaceutical Technology, Faculty of Pharmacy, Masaryk University, 61200 Brno, Czech Republic; 507072@muni.cz (T.B.); vyslouzilj@pharm.muni.cz (J.V.); simickovama@gmail.com (M.N.); muselikj@pharm.muni.cz (J.M.); pavelkovam@pharm.muni.cz (M.P.); draha.zednickova@seznam.cz (D.Č.); 2Department of Pharmacology and Toxicology, Veterinary Research Institute, 62100 Brno, Czech Republic; masek@vri.cz; 3Department of Chemistry, Faculty of Pharmacy, Masaryk University, 61200 Brno, Czech Republic; culenm@pharm.muni.cz

**Keywords:** matrix tablets, HPMC, Neusilin^®^ US2, microcrystalline cellulose, USP apparatus 2 dissolution test, dynamic dissolution study

## Abstract

Modern pharmaceutical technology still seeks new excipients and investigates the further use in already known ones. An example is magnesium aluminometasilicate Neusilin^®^ US2 (NEU), a commonly used inert filler with unique properties that are usable in various pharmaceutical fields of interest. We aimed to explore its application in hypromellose matrix systems (HPMC content 10–30%) compared to the traditionally used microcrystalline cellulose (MCC) PH 102. The properties of powder mixtures and directly compressed tablets containing individual fillers NEU or MCC, or their blend with ratios of 1.5:1, 1:1, and 0.5:1 were investigated. Besides the routine pharmaceutical testing, we have enriched the matrices’ evaluation with a biorelevant dynamic dissolution study and advanced statistical analysis. Under the USP apparatus 2 dissolution test, NEU, individually, did not provide advantages compared to MCC. The primary limitations were the *burst effect* increase followed by faster drug release at the 10–20% HPMC concentrations. However, the biorelevant dynamic dissolution study did not confirm these findings and showed similarities in dissolution profiles. It indicates the limitations of pharmacopoeial methods in matrix tablet development. Surprisingly, the NEU/MCC blend matrices at the same HPMC concentration showed technologically advantageous properties. Besides improved flowability, tablet hardness, and a positive impact on the in vitro drug dissolution profile toward zero-order kinetics, the USP 2 dissolution data of the samples N75M50 and N50M50 showed a similarity to those obtained from the dynamic biorelevant apparatus with multi-compartment structure. This finding demonstrates the more predictable in vivo behaviour of the developed matrix systems in human organisms.

## 1. Introduction

Hydrophilic matrices are the most widely used sustained-release oral dosage forms. Their advantages lie primarily in the flexibility of dissolution behaviour, the low cost of the tableting process, and the availability of numerous swelling polymeric carriers differing in their origin [1]. Semisynthetic polymer hypromellose (HPMC) is the first-choice pharmaceutical excipient with a unique position in the field of matrix systems. Its advantages include its safety profile, biocompatibility, flexibility, stability within a wide pH range, resistance to enzymatic cleavage, and predictable in vivo behaviour [2]. It is available under several trademarks, and its linear polysaccharide molecules differ in the degree of substitution, ratio of side groups, final molecular weight, and solution viscosity. These parameters significantly affect the drug’s liberation from HPMC-based matrix systems and influence standard processes such as surface hydration, swelling, and gel layer formation [3]. In combination with API and other pharmaceutical excipients, HPMC co-participates in the dissolution behaviour of the final matrix system, characterized by the drug’s dissolution rate, kinetics, and release mechanism, including the diffusion of API molecules through the surface gel layer, erosion, or a combination of both processes [4]. The thickness of the gel layer and its rigidity are the primary criteria influencing the *burst effect* typical for hydrophilic matrix systems and, overall, the whole dissolution profile [5].

Insoluble filler microcrystalline cellulose (MCC) is a commonly used filler in matrix tablets technology. It exhibits an excellent densification and formation of tablets with optimal mechanical properties. It is known under several trademarks (Avicel^®^, Ceolus^®^, Emcocel^®^, Vivapur^®^, etc.). Its porous particles possess a specific surface area of ~1.18 m^2^/g [6], given by randomly clustered fibrous microcrystals. MCC particles differ in their particle size (20–200 µm) and humidity content (1.5–5%) [7]. Its increasing concentration in HPMC matrices ensures a faster drug release profile and modulates drug release behaviour according to requirements [8]. Despite its insolubility, MCC shows a limited ability to swell. Compared to soluble lactose, MCC, combined with HPMC, contributes to the polymer swelling and supports a higher rigidity of the surface gel layer, resulting in more prolonged drug release [9]. On the other hand, insoluble dibasic calcium phosphate was recognized as a more efficient retardant than MCC PH 101 for a slightly soluble drug from the HPMC systems due to its hydrophobic character [10].

Magnesium aluminosilicates (MAS) (Neuslin^®^, Pharmsorb^®^, Veegum^®^) are an exceptional group of pharmaceutical excipients with excellent potential for the matrix tablets technology. They are available in numerous types as neutral or alkaline substances, and they become more and more perspective in modern drug delivery systems [11]. In the silicate family, Neusilin^®^ is an entirely synthetic MAS, insoluble in water, and available in 11 different grades as an amorphous white powder or granules with the empirical formula Al_2_O_3_·MgO·1.7SiO_2_·xH_2_O. It is prepared using the spray drying process, resulting in an extensive specific surface area (100–300 m^2^/g) [12], high absorption capacity (up to 3.4 mL/g), high porosity [13], and good flow [14] and compression properties [15,16]. In an aqueous medium, it does not form a gel [12]. In modern pharmaceutical development, Neusilin^®^ is used mainly in the technology of liquisolid systems [17], self-emulsifying systems [18,19,20], enzyme immobilization [21], solid dispersions [22], and others.

Information about MAS’s role in the traditional technology of matrix tablets can be rarely found in the scientific literature. The excellent absorption capacity of Neusilin^®^ US2 was used in the application of Eudragit^®^ water dispersions on a mixture of a freely soluble drug and Neusilin^®^ to reduce the *burst effect* of HPMC matrices [23]. In the advanced technology of HPMC matrices, propranolol–MAS intercalated complexes as drug reservoirs were incorporated into HPMC tablets to achieve zero-order kinetics [24]. Nevertheless, a study focusing on Neusilin^®^ behaviour in the function of a traditional filler in matrix systems has not been published.

Therefore, the objective of this experimental work was to profoundly investigate whether Neusilin^®^ US2 is a suitable filler excipient for HPMC matrix systems and if it could be an acceptable alternative to the widely used MCC. Matrix systems were explored mainly by an in vitro pharmacopeial dissolution test, an advanced dynamic dissolution study, a gel layers evaluation, cryo-SEM, and multivariate data analysis.

## 2. Materials and Methods

### 2.1. Materials

Caffeine (Zentiva k.s, Prague, Czech Republic) was selected as a slightly soluble model drug. Hypromellose—HPMC K4M (Colorcon Limited, Dartford, UK) was a release-retarding polymeric carrier. 

Magnesium aluminometasilicate Neusilin^®^ US2 (Fuji Chemical Industries Co., Ltd., Toyoma, Japan) and microcrystalline cellulose Avicel^®^ PH 102 (MCC) (FMC Biopolymers, Rockland, ME, USA) were added as the compared insoluble fillers. Magnesium stearate (Peter Greven, Bad Münstereifel, Germany) and colloidal SiO_2_ (Aerosil^®^ 200) (Degussa, Vicenza, IT) were used to facilitate the powder blends flow. The chemicals used for the preparation of dissolution media were as follows: for USP 2 dissolution test—sodium chloride; hydrochloric acid (1 M) for preparation of 1.2 pH artificial gastric juice without pepsin (2 g/80 mL per 1000 g); sodium triphosphate for its pH adjustment to pH 6.8; for dynamic dissolution test—sodium chloride; potassium chloride; and pepsin (all Sigma–Aldrich, St. Louis, MO, USA) for preparation of the pH 1.8 dissolution medium (4 g/0.2 g/2.6 g per 2000 g). For HPLC, a mixture of acetic acid (Dr. Kulich Pharma, Hradec Králové, Czech Republic) and methanol (Honeywell, Bucharest, RO) in an 80:20 ratio was used as a mobile phase.

### 2.2. Preparation and Evaluation of Powder Blends for Direct Compression and Matrix Tablets

All ingredients in amounts according to Table 1 were mixed using a 3-axial homogenizer Turbula (T2C WAB, Basel, CH) for 10 min. Before the matrix tablets’ preparation, the flow properties of the powder blends were evaluated accordingly to Eur. Ph. 10, namely Hausner’s ratio (n = 3, SVM 102, Erweka, Heusenstamm, Germany). Convex-faced HPMC matrices (ø 10 mm, 258 mg) were prepared by direct compression using eccentric tablet press (Korsch EK 0, Korsch Pressen, Berlin, Germany) to maximum hardness. The pharmacopeial evaluation (Ph. Eur. 10) included the prescribed tests, weight (n = 20) and content (n = 10), uniformity, hardness (n = 10), and friability (6.5 g, 4 min., 25 rpm).

### 2.3. Scanning Electron Microscopy (SEM)

The surface topography of matrix tablets was analysed using scanning electron microscopy (SEM). Images of the platinum-coated (layer thickness-10 nm) samples were taken using a MIRA3 scanning electron microscope (Tescan Brno, Brno, Czech Republic) at an accelerating voltage of 5.0 kV and 1.97 and 1.99 kx magnification.

### 2.4. Pharmacopeial Dissolution Study Using USP Apparatus 2

The USP apparatus 2 (paddle) was employed to obtain 12 h dissolution profiles (SOTAX AT 7 Online System, Donau Lab, Zurich, CH) at 50 rpm and 37 °C. To approximate the natural pH GIT conditions, the dissolution started in 900 mL of pH 1.2 artificial gastric juice without pepsin. After 2 h, the pH was increased to 6.8 by 18.7 g of sodium triphosphate per vessel [25]. Automatic sampling was pre-set at 30 min, 60 min, and then every hour for the total of the 12-h dissolution profile. Tablets were placed in sinkers to avoid tablet adhesion and/or flotation. Automated UV/Vis analysis using a 275-nanometre maximum was used for the drug released amounts. For each sample, six units were measured. The mean values and the standard deviations (SD) were calculated for each sample in all time points.

### 2.5. Biorelevant Dissolution Study Using Golem^®^ v2 Apparatus

Biorelevant dissolution was performed in the Golem v2 apparatus (Figure 1). It is a partially computer-controlled artificial GIT simulator designed for testing solid dosage forms for oral administration. It consists of a heated plate, where the following four compartments are placed: stomach, duodenum, jejunum, and ileum. Each compartment is made from two-layer (polyolefin/polyamide) intravenous bags (Baxter Viaflo, Baxter Healthcare Ltd., Thetford, UK) that were modified by welding to provide a specific inner geometry and incorporated a pH probe [26]. Compartments are connected through peristaltic pumps, which enable the movement of predetermined volumes between consecutive compartments. On each compartment, a paddle is placed to provide mechanical force for mixing. To maintain physiological conditions during the experiment, it is possible to adjust and measure the pH and adjust volumes, transit times, the temperature, and the mixing speed of the paddles in the range of 1–7 lifts per minute.

The dissolution tests were performed at 37 °C three times for every sample, with five paddle lifts per minute for all compartments. The starting volume was 231 mL [27,28] of pH 1.8 dissolution medium in the stomach compartment (fasted state with a glass of water) [29]. During the dissolution, the pH in the compartments was maintained via the addition of either 1 M HCl solution or 1 M NaOH + 0.24 M NaHCO_3_: duodenum at pH 5.0 [30], jejunum at pH 6.0 [31], and ileum at pH 7.0 [32]. The whole experiment ran based on a template created explicitly for Golem v2 software. Visualization of volume movements and sampling points can be seen in Figure 2. As the system itself cannot transport tablets from one compartment to another, the sinker (27 mm in length and 11.6 mm in width) equipped with a string was used for this purpose (Figure 3). The sinker was gradually placed into the stomach, duodenum, jejunum, and ileum compartments for 30, 10 [33], 80, and 120 min, respectively, to acquire 240-min dissolution profiles [34]. Samples were collected using 10-millilitre syringes and a 0.2-micrometre nylon syringe filter and analysed using HPLC apparatus Agilent 1250 Infinity (Agilent Technologies, Waldbronn, Germany). The chromatographic separation was performed using Supercosil ABZ + Plus column (150 mm × 4.6 mm × 3 µm, Sigma Aldrich, Bellafonte, PA, USA). The mobile phase consisted of methanol and a 50 mM acetate buffer (20:80, *v*/*v*) adjusted to pH 4.0. Other measurement conditions were as follows: a flow rate of 1.5 mL/min, an injection volume of 10 µL, and a column temperature of 40 °C. The spectra were recorded in the wavelength range between 190 and 400 nm. The chromatogram was acquired at 294 nm [26].

### 2.6. Similarity Factor Analysis and Drug Release Kinetics

To compare the caffeine dissolution profiles from both dissolution studies, similarity factors *f*_2_ were calculated according to the well-known equation [35]. The calculation of *f*_2_ factors between corresponding samples was used to observe the influence of the fillers (MCC PH 102 and Neusilin^®^ US2) on dissolution characteristics within individual dissolution methods and between them. Additionally, drug release mechanisms and its kinetics were investigated. Gathered dissolution data were treated with known mathematical models (Higuchi equation, Korsmeyer–Peppas equation, zero-order equation, first-order equation) to find a correlation using a non-linear regression method [35].

### 2.7. Dynamic Characteristics of the Gel Layer

The gel layer thickness and penetration force were investigated to describe polymer swelling characteristics. Penetration force was expressed as performed work in time. Measured tablets were treated under the same conditions as the pharmacopeial dissolution test. Measurements were performed after 15 min, 30 min, and then every 30 min until the final 360 min, using Texture Analyser CT 3 (Brookfield, London, UK). For each sample, three tablets were placed into a polyvinylchloride (PVC) mould, allowing only upper surface swelling [36]. At the sampling time, the mould with the tablet was secured to the centre of the texture analyser platform. A 2-millimetre diameter/30-millimetre length rod probe TA39 was used for measurement, with an approaching speed of 0.5 mm/s. The actual measurement started after a 5-g trigger load was reached (corresponding to the probe/gel surface contact), with a 250-g trigger load as the termination limit (corresponding to the probe hitting non-hydrated structures) [37]. Measured data were processed using Texture Expert software (Brookfield, London, UK). For each sample and both parameters, the mean values and the SDs were calculated from three units in each sampling point.

### 2.8. Characterization of Gel Layer by Scanning Electron Cryomicroscopy (CryoSEM)

The surface morphology of the gel layer was studied using a scanning electron cryomicroscopy technique (SU 8010, Hitachi, Tokyo, Japan). Observed tablets were treated in the same conditions as in the pharmacopeial dissolution test for three hours, using the PVC moulds to restrict the swelling to the upper surface only. After removal, the tablets were transversely cut in half and frozen in liquid nitrogen. Such prepared samples were moved to the cryomicroscopy preparation chamber. Samples were coated by Pt/Pd, following surface water/ice removal by sublimation. Observed cross-sections were kept at −130 °C.

### 2.9. Multivariate Data Analysis

For the evaluation of formulation variables (the content of HPMC, NEU, and MCC) and their interaction with qualitative parameters of the powder blends and matrix tablets (flowability; tablet hardness; amount of caffeine released at 30 min, 240 min, and 360 min; gel layer thickness at 360 min; penetration force at 15 min and 360 min), factor analysis with factor rotation (varimax normalized) was used. Before the evaluation, the data were standardized to zero column mean and unit variance to reduce the effect of variables of different units. Moreover, a statistical analysis was also performed to evaluate the impact of the dissolution method on drug release characteristics (USP apparatus 2 vs. dynamic dissolution study). The evaluation was performed using the Statistica 12 program (StatSoft. Inc., Tulsa, OK, USA). Verification of the presence of remote points and verification of multidimensional normality was performed using the program QC.Expert 3.2 (TriloByteStatistical Software, Pardubice, Czech Republic). 

## 3. Results and Discussion

In this experimental study, MCC PH 102, Neusilin^®^ US2, and their mixture were used in HPMC matrices and thoroughly compared to investigate if Neusilin^®^ US2 (NEU) is an acceptable alternative to the widely used MCC. The basic properties of the powder blends and the tablets are displayed in Table 2. According to the Ph. Eur. 10, the flow characteristics of the powder blends were evaluated as passable or poor (HR 1.28–1.39). The results expectedly indicated a positive influence of NEU with spherical particles [15,38] compared to fibrous structured MCC [7]. A deterioration of the ability to flow was noticed at higher HPMC concentrations in the powder mixtures due to the polymer’s smaller and irregularly shaped particles [39].

The matrix tablets with an average weight of 250.1–261.7 mg were compressed to maximum hardness (98.2–156.0 N). Friability reached satisfactorily low values of 0.13–0.30%, and all the matrix tablets corresponded to Ph. Eur. 10 limits. The highest hardness was found for samples containing both MCC and NEU together. As stated in the scientific literature, NEU as a filler exhibits excellent compression properties, and its addition to the tableting mixture allows for the hardness of the final tablets to be improved [40]. Compared to the MCC samples, the matrix tablets with only NEU possessed a relatively lower hardness, similarly as in the experimental study conducted by Adebisi et al. (2015) [41]. Conforming to these findings, the SEM images (Figure 4) show the more compact surface of sample N75M25 than sample M125 and sample N125 with a more furrowed surface.

### 3.1. Pharmacopeial Dissolution Study Using USP Apparatus 2, Similarity Factor Analysis, Drug Release Kinetics, Multivariate Data Analysis

At first, the in vitro dissolution profiles of the formulated matrix tablets obtained under the conditions of the USP apparatus 2 method were consistent with the statement that increasing the polymer:filler ratio reliably reduces the permeability of the surface gel layer controlling the drug release due to an increase in its viscosity and rigidity [42].

Although soluble fillers (e.g., lactose, sucrose) have a little more visible effect on the drug release compared to insoluble ones, a significant influence of used insoluble filler/s on the samples’ dissolution behaviour was observed due to the relatively low HPMC content (10–30%) [43]. The obtained dissolution profiles are depicted in Figure 5; the determination coefficients of the mathematical models are summarized in Table 3 and similarity factors in Table 4.

The first set of samples containing approximately 10% HPMC K4M (M125, N125, N75M50) could generally not achieve controlled drug release due to the low polymer concentration [44]. However, the sample containing only MCC as a filler (sample M125) showed a prolonged drug release (*burst effect*, 17.11%; totally, 75.97%; over 12 h). Sample N125 released 42.33 and 81.24% of the drug during the first 30 min and the whole dissolution test, respectively, by completely replacing the MCC with NEU. The sample with the mixed insoluble fillers N75M50 released only 14.35% of the drug in the first half an hour and the highest amount of 78.37% totally. It is evident that the combination of 10% HPMC with NEU significantly accelerated the drug release from the matrix system (*burst effect* increased by 59.6%) and led to a significant change in the dissolution profile (*f*_2_ 37.94) in comparison with the MCC only sample. On the other hand, a partial replacement of MCC in sample N75M50 resulted in a slight reduction in the initial drug release rate and a slight increase in the total drug release during the 12-h dissolution course. The dissolution data shift towards zero-order kinetics was observed (see Table 3, *R*^2^ 0.6066) while maintaining a similarity between dissolution profiles (*f*_2_ 73.05).

In the set of samples with the 20% HPMC K4M amount (M100, N100, N50M50), an increase in the polymer:filler ratio expectedly led to a decrease in both the *burst effect* (11.99–18.97%) and the total drug release (57.67–69.01%) [45]. The drug amount of 57.42% was released from the M100 sample during the dissolution test, and its *burst effect* reached 12.96%. A replacement of the MCC filler with NEU (N100 sample) resulted in a significantly faster drug release (*f*_2_ 45.96) throughout the dissolution course with a *burst effect* of 18.79%. The comparison of the dissolution profiles of the M100 and N50M50 samples is also worth mentioning. Despite the similarity of their dissolution curves (*f*_2_ 69.52), the obtained profiles are almost identical within 420 min, while they become diverging after that. The sample N50M50 released 14.31% of the drug until the end of the dissolution test, whereas the sample M100 released only 4.80%. Again, a phenomenon positively affecting the dissolution profile toward zero-order kinetics (see Table 2, N50M50: *R*^2^ 0.6602; M100: *R*^2^ 0.1304) was evidently observed, i.e., a slight decrease in the *burst effect* and a more pronounced increase in the total drug released during the 12-h dissolution test.

In the sample set with the maximum 30% HPMC K4M concentration, the overall lowest *burst effect* (9.2–11.47%) and the smallest amount of drug released during the 12-h dissolution test (45.70–57.49%) were expected. All the samples with the highest HPMC concentration were revealed as similar (*f*_2_ 61.95–83.63). Unlike the previously evaluated sets, the type of filler used did not significantly affect the in vitro dissolution behaviour of the matrix systems. The impact of the HPMC prevailed, and the previously substantial influence of the fillers was not manifested [46].

Figure 2 shows a similar but milder attribute as described above, and in the second half of the dissolution study, the sample N25M50 exceeded the samples M75 and N75 in the amount of the drug released (N25M50—11.59% vs. M75—7.00% vs. N75—2.52% of the drug). It also exhibited the highest value of *R*^2^ for zero-order kinetics of this subset (see Table 2, *R*^2^ 0.5366). Moreover, it can be noticed that at a 20 or 30% HPMC level, the matrices containing MCC or NEU as fillers exhibited a significant decrease in the drug release rate during dissolution course by supporting the immobility of slightly soluble drug particles in swollen matrices [42].

The caffeine release mechanics from the HPMC matrices under USP II conditions can be estimated from release exponent *n* (Korsmeyer–Peppas model—Table 3). The release exponent *n* values were taken for comparison only in a case of a good match with the kinetic model (*R*^2^ ≥ 0.9924). It ranges between 0.485 and 0.568 and points out anomalous transport, including the diffusion and erosion processes’ participation in the drug release. Based on its value, the NEU samples (*n* = 0.568 for N100, *R*^2^ = 0.9924) showed the erosion of the gel layer to a higher extent compared to MCC only (*n* = 0.485–0.515). Following the previously mentioned statement, MCC can partially swell contrary to NEU, which disturbs the gel layer’s integrity more and thus backs matrix erosion. Release exponent *n* of the mixed-filler samples shows that a decrease in the NEU/HPMC ratio weakened the role of the erosion process in caffeine release from HPMC matrices.

### 3.2. Investigation of Gel Layer Characteristics (USP Apparatus 2)

The course of the gel layer formation is displayed in Figure 6. Due to the low HMPC content, the slight swelling of the matrix surface and the narrowest gel layer were observed in the 10% HPMC samples (M125, N125, N75M50). These results are consistent with the claim that a higher gel-forming polymer content leads to the formation of a wider gel layer on the matrix surface, more effectively reducing the drug amount released in time [47].

Comparing the samples with 10 and 20% HPMC, the proportion of HPMC (the matrix filler amount: 50 and 40%, respectively), a broader gel layer was formed in the NEU matrix tablets after 15 min compared to the corresponding MCC samples (0.28 mm for M125, 0.37 mm for N 125, 0.33 mm for M100, and 0.41 mm for N100). Still, a lower penetration force was needed to overcome it (Figure 4, Table 5). This finding is probably related to the higher adsorption capacity of the NEU as a filler [16]. Due to this, an extensive transport of the dissolution medium to the tablet structure [15] probably supported an HPMC swelling extent. However, the individual NEU particles disrupted the compactness of the forming gel layer more significantly than the MCC or the NEU/MCC combination, resulting in an acceleration of the initial drug release [44]. Figure 7 shows the cryo-SEM images depicting the selected swollen tablets’ gel layer in the 30th minute of the dissolution test. A decrease in the gel layer’s compactness can be clearly seen in order: M50N50, M100, N100, and it agrees with the *burst effect* extent.

The gel layer growth between the 15th and 360th minute reached values of 0.75 mm for M125, 0.57 mm for N125, 0.78 mm for M100, and 0.61 mm for N100. It can be stated that the swelling extent was more intensive for the MCC matrix tablets compared to the NEU ones. A slight decrease in the penetration force in the given time interval was recorded for the MCC samples, and an opposite trend was visible in the NEU matrices. Due to the ability to swell and promote water absorption over a long period, MCC disturbs the integrity of the gel layer of HPMC tablets less, for which a reduction in the rigidity of the gel layer over time is usual [48]. The increase in the penetration force of NEU samples can be attributed to the lower hydration of the insoluble NEU particles isolated from the gel structure and corresponds to a lower swelling extent over time. These results are consistent with the total drug released during the dissolution study.

A different situation was revealed for the 30% HPMC matrix tablets. The effect of the selected filler (MCC or NEU, 30% per tablet) on the thickness of the gel layer (the difference was only 0.06 mm) was limited by the higher proportion of HPMC in the matrix, leading to a formation of a gel layer with a significantly higher viscosity. The difference in the *burst effect* of the samples M75 and N75 was only 2.3% in the 15th minute. The lowest total amount was released from the sample N75 (45.7%), probably due to the drug’s more difficult release from the porous NEU’s internal structures through the highly viscous gel layer [49].

The samples with the MCC/NEU combination exhibited gel layers of the comparable thickness such as the samples with MCC only (0.28–0.38 mm for MCC samples vs. 0.28–0.40 mm for the filler blends) and rigidity in 15 min (penetration force 0.69–0.72 N for MCC samples vs. 0.68–0.76 N for the filler blends), see Table 5 and Figure 6. The results are consistent with the values found for the drug released at the beginning of the dissolution test, which did not differ by more than 2.8%. The samples with the filler’s combination in cooperation with the HPMC polymer were optimal for forming a quality gel layer and always showed a slight decrease in the *burst effect*, which was more pronounced at lower HPMC concentrations in the tablet. The increase in the gel layer in the interval between the 15th and 360th minutes reached values of 0.75 mm for M125, 0.53 mm for N75M50, 0.78 mm for M100, 0.61 mm for N50M50, 1.00 mm for M75, and 0.76 mm for N25M50. The data showed a negative effect of the NEU presence on the HPMC swelling ability and corresponded to a more significant amount of drug released from samples in the given time interval.

### 3.3. Multivariate Data Analysis (USP Apparatus 2)

The data were evaluated using factor analysis to find the relationships between the formulation variables (the proportion of used pharmaceutical excipients—HPMC, NEU, MCC) and the following selected quality parameters of the tablets: in vitro dissolution profile (USP II) parameters, namely, *DIS30*, *DIS240*, and *DIS720* (amount of drug released at given time point); the gel layer parameters, namely, thickness at the 360th min, and penetration force at the 15th and 360th minutes, together with tablet hardness. The created PCA model is displayed in Figure 5. The data obtained by the dynamic dissolution test (*Golem 30* and *Golem 240*) will be discussed later.

In the factor loadings graph (Figure 8), the first factor is explained by the content of MCC or NEU in the mixture and the second one by the HPMC content. It shows that increasing the HPMC matrix’s MCC:NEU ratio positively influences tablet hardness, gives the gel layer resistance at the beginning of the dissolution test (MCC position near *Penetration force 15*), and reduces the initial *burst effect* (*DIS30* position is in the opposite segment). The increasing proportion of NEU in the matrix system ensures a higher rigidity of the gel layer in the later stages of the dissolution test (NEU position near *Penetration force 360*). The increasing proportion of HPMC (factor two) in the matrix positively affects tablet swelling (HPMC near *Thickness 360*). In contrast, its increasing concentration reduces the total drug released during the 12-h dissolution test (HPMC and *DIS720* in the opposite segment). As shown from the above PCA analysis, MCC and NEU are not freely interchangeable in the matrix tablet technology based on USP II conditions, but the strong dependence of the HPMC concentration exists in the system. While the 30% HPMC in tablets does not bring a significant difference, the filler selection is crucial at its lower concentration. According to these results, a combination of MCC and NEU in matrix tablets with a 10% HPMC concentration (sample N75M50) and especially with a 20% HPMC concentration (sample N50M50) can be advantageously used in the adjustment of the drug release profile towards the zero-order kinetic.

The above-described claims remained to be verified by a biorelevant dissolution study using apparatus.

### 3.4. Biorelevant Dissolution Study

Classic USP apparatuses (Eur. Ph.—Paddle method) using a minimum of 500 mL of dissolution medium are the first choice in pharmaceutical development and quality control. The limitations coming from the difference between the artificial system and the real digestive system can be partially overcome using biorelevant media [50]. Nevertheless, it is still challenging to simulate the complexity of in vivo conditions in in vitro devices. Therefore, for the greater potential of correlation with in vivo drug release behaviour, biorelevant dissolution devices come to assist the standard dissolution tests. They are designed around the idea of physiological simulation conditions that are as accurate as possible to mimic the influences drugs are exposed to during passage through the GI tract. These include ever-changing pH values, medium composition, dynamic volume changes, transition and residence times, and definable mechanical stress on the dosage form as the critical factors [51]. For that reason, to confirm the trends observed in the prepared matrix tablets, a dynamic dissolution study mimicking the GIT fasted state was performed using the Golem^®^ v2 apparatus. It is a dynamic four-compartmental apparatus simulating chyme transit and biorelevant conditions in four GIT compartments, i.e., the stomach, duodenum, jejunum, and ileum. Biorelevant dynamic dissolution study was focused on the drug release behaviour during the initial 4 h of dissolution, on which, as a rule, the entire course of drug release depends most. This study should yield profiles closer to the in vivo reality, as the device enables a more accurate simulation of the GIT [52].

In the design of the dynamic biorelevant study, emphasis was placed on the physiological volumes (much lower for the fasted state compared to the USP apparatus II), thorough pH control across all the compartments, and on the dynamic element of the dissolution itself (which is completely unattainable in standard USP apparatus I/II). The obtained results of the dynamic dissolution test are depicted in Figure 9, and the determination coefficients *R*^2^ of the dissolution data fitting to the used mathematical models are shown in Table 6. First and foremost, it must be mentioned that in multivariate data analysis, the chosen sampling points from biorelevant dissolution profiles remained in close vicinity to their complimentary USP dissolution sampling points (see Figure 5). It means that the observed relationships between the excipients’ influence and the observed parameters should remain the same for both dissolution methods. Therefore, theoretically, they could also be translated to in vivo as well.

Nevertheless, it is evident that the dissolution performance under the biorelevant conditions was visibly slower, *burst effects* were reduced, and, in addition, the profiles were tighter. It can be explained by the lower but more physiological compartment volumes used in the dynamic biorelevant dissolution study. For example, the fastest-releasing sample under the USP II conditions, the N125 sample with 72.6% released in 4 h, was reduced to 45.6% at the same time. Moreover, in contrast to the USP II conditions, all the complimentary samples were similar. This observation could hint at the possible mitigation or even negation of the filler effect on the dissolution profile under conditions closer to a physiological state.

Unlike other formulations, a plateau between the 22nd and 33rd minutes was observed for the N125 sample. This effect could probably be caused by the highest content of NEU filler with a tremendous absorption capacity. After the drug is released from the tablet’s surface, the flow of the dissolution medium into NEU particles probably prevailed for a short time, followed by an exponential increase in drug release due to a limited gel layer formation in a given interval, which is also noticeable for sample N125 from Figure 6.

Comparing the obtained data for complimentary samples from Table 2 and Table 6 (taken value for *R*^2^ ≥ 0.9925), the release exponent *n* from the Korsmeyer–Peppas equation was significantly higher under biorelevant (0.64–0.73) than the USP II conditions (0.485–0.568). It suggests a decreased diffusion participation in drug liberation on behalf of gel layer erosion. It can be explained by a lower (but more physiological) volume, resulting in a reduction in the drug concentration gradient and a more physiological type of deformation. A less pronounced influence of the used filler on the release mechanism was found out (MCC samples—*n* = 0.64–0.70; NEU samples—*n* = 0.72–073), which supports the filler effect mitigation on drug release under the dynamic dissolution test stated above. The exploration of kinetic data from biorelevant conditions also showed the desired shift to zero-order kinetics (*R*^2^ ≥ 0.845).

The difference between the profiles obtained via the USP II and biorelevant dissolution apparatus can also be observed through similarity factor analysis (Table 4). It reveals that the profiles are not similar in all the cases except for the 10–20% HPMC samples with a filler combination. In both N75M50/N75M50 and N50M50/N50M50, respectively, the *f*_2_ values exceeded the value of 50 (58.66 and 55.45, respectively). This finding suggests that the samples prepared with such a combination of excipients could have the best predictability of in vivo performance based on its dissolution profile.

This part of the study also showed that it could be imperative to employ dissolution devices for controlled release dosage forms to imitate the in vivo conditions better than standard USP apparatuses I and II, at least in the form of a reciprocating cylinder or flow-through cells, which are the most often recommended devices for assessing modified-release formulations [53,54]. However, these do not meet the conditions of bio relevance to such an extent as dynamic dissolution devices.

## 4. Conclusions

In this experimental study, the fillers NEU, MCC, and their combination were investigated as a part of HPMC systems. While the USP results showed that the NEU was unsuitable for formulating HPMC matrices due to much faster drug release, the biorelevant dynamic dissolution test did not confirm these findings and partially mitigated the differences between the filler’s influence. This observation suggests that the dynamic biorelevant dissolution methods could very well serve as a valuable tool to complete the dissolution image. The fascinating finding was made at the 10–20% HPMC matrix systems in the case of the NEU/MCC combination samples. Not only did the addition of NEU not accelerate the release in comparison with the MCC-only sample, but it actually slightly slowed the drug release and shifted the profile closer to zero-order kinetics, even beyond the performance of the MCC-only samples. The comparison of the USP dissolution and biorelevant method through similarity factor analysis also revealed that only the profiles of the NEU/MCC samples possess a similarity. It suggests that the NEU/MCC filler combination could exhibit better predictability of the in vivo behaviour in the matrix tablets technology, which could be valuable in developing matrix or self-emulsifying systems. Additionally, it once again hints at an advantageous combined use of multiple distinct dissolution methods to boost the discriminant ability of the dissolution testing. Further study is planned to investigate the different NEU/MCC ratios to eventually find the boundaries of the observed synergy with the eventual involvement of the in vivo study to correlate with dissolution analysis.

## Figures and Tables

**Figure 1 pharmaceutics-14-00127-f001:**
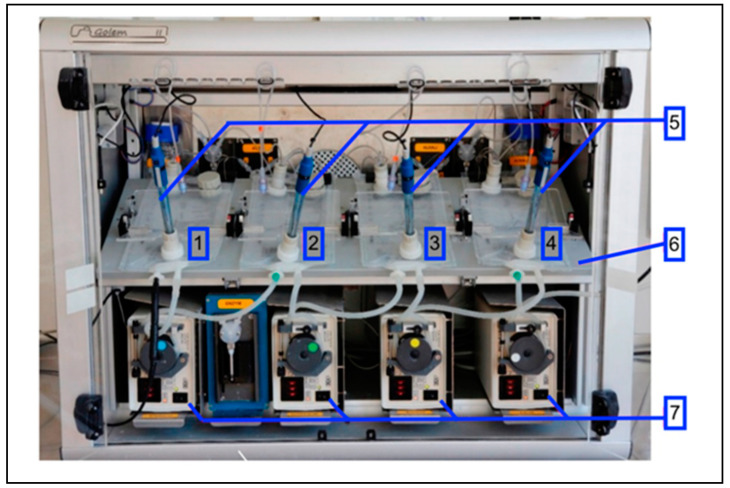
Golem v2 apparatus from front view: (1) stomach, (2) duodenum, (3) jejunum, (4) ileum, (5) pH probes, (6) heating platform, (7) peristaltic pumps.

**Figure 2 pharmaceutics-14-00127-f002:**
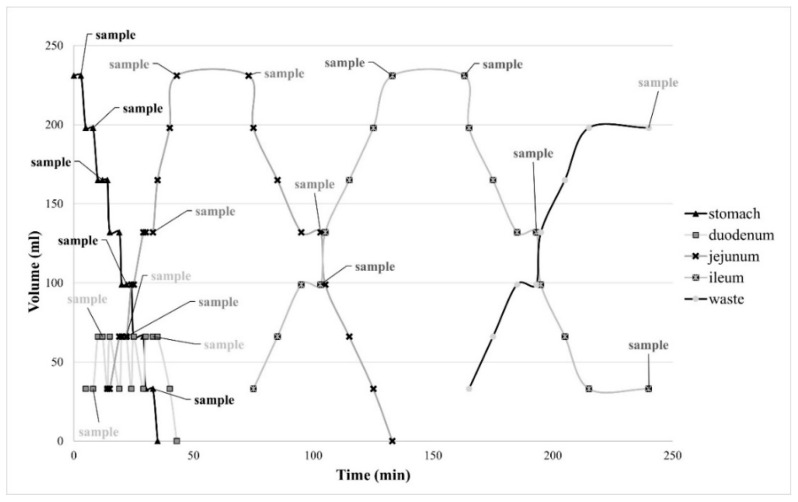
Template diagram with predetermined volume movement and sampling points.

**Figure 3 pharmaceutics-14-00127-f003:**
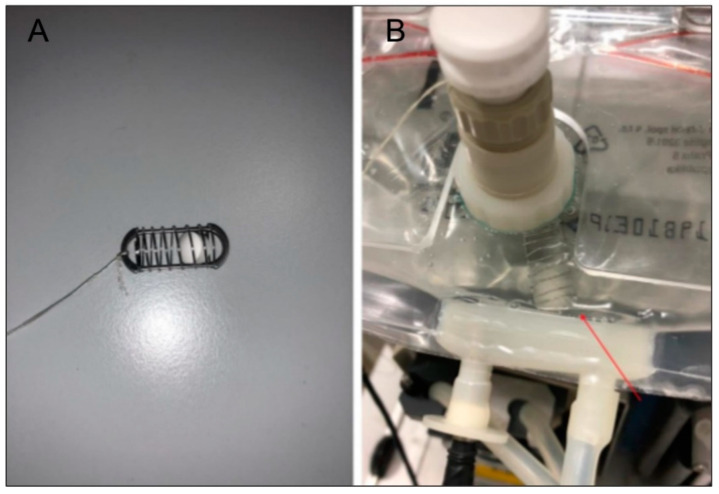
Sinker with incorporated tablet (**A**), sinker with a tablet inside of the bag (**B**).

**Figure 4 pharmaceutics-14-00127-f004:**
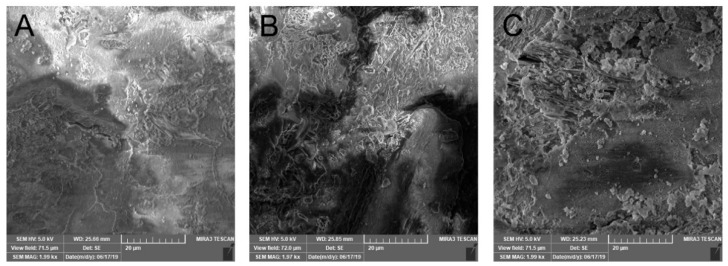
SEM surface topography of the selected samples of matrix tablets: (**A**) N75M50, (**B**) M125, (**C**) N125.

**Figure 5 pharmaceutics-14-00127-f005:**
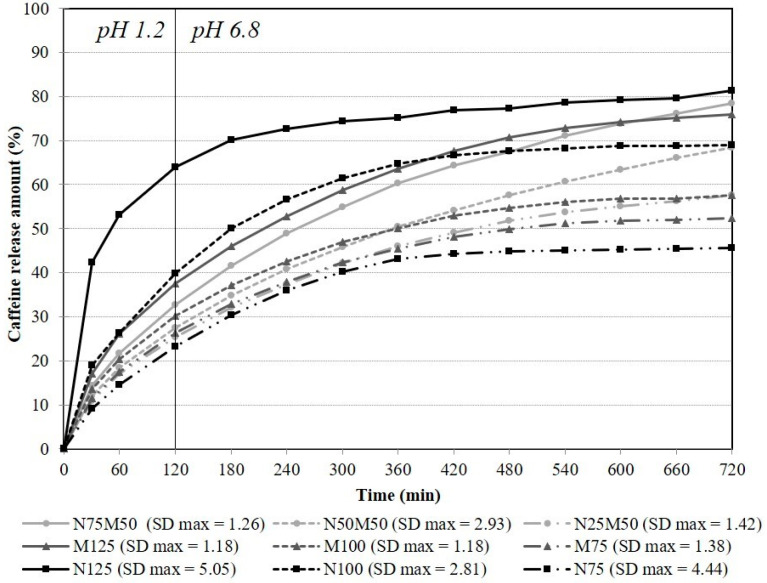
In vitro dissolution behaviour of the matrix tablets under USP apparatus 2 conditions.

**Figure 6 pharmaceutics-14-00127-f006:**
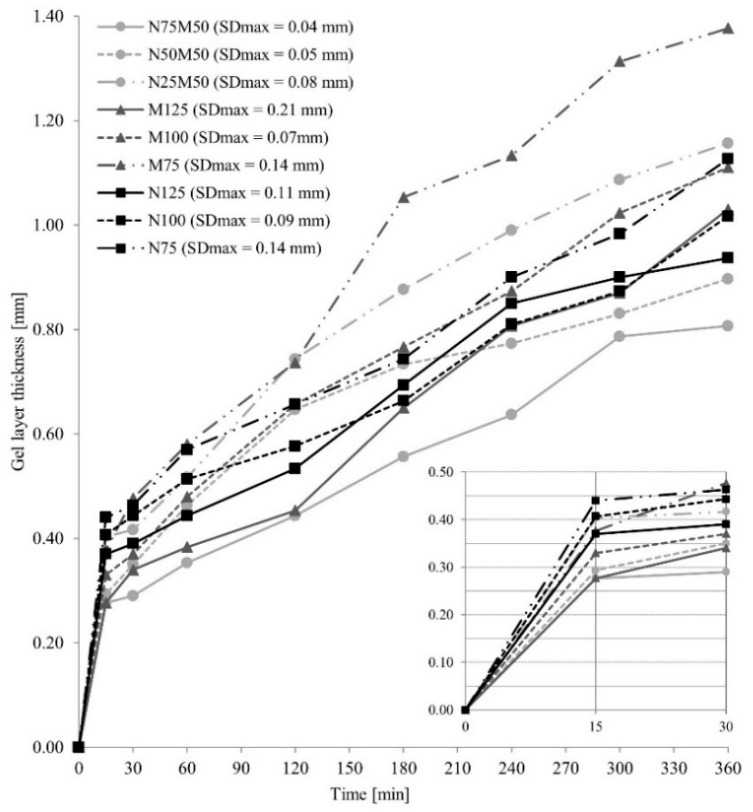
The gel layer thickness of the HPMC matric tablets.

**Figure 7 pharmaceutics-14-00127-f007:**
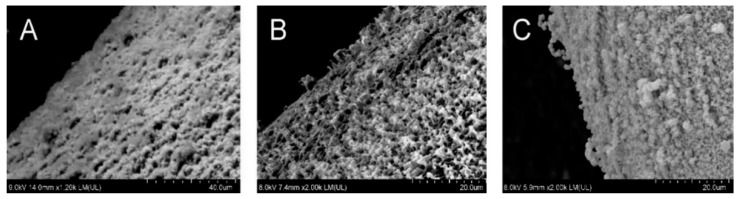
Cryo-SEM images showing gel layer of swollen tablet samples in the 30th min of the dissolution test: (**A**) M100, (**B**) N100, (**C**) M50N50.

**Figure 8 pharmaceutics-14-00127-f008:**
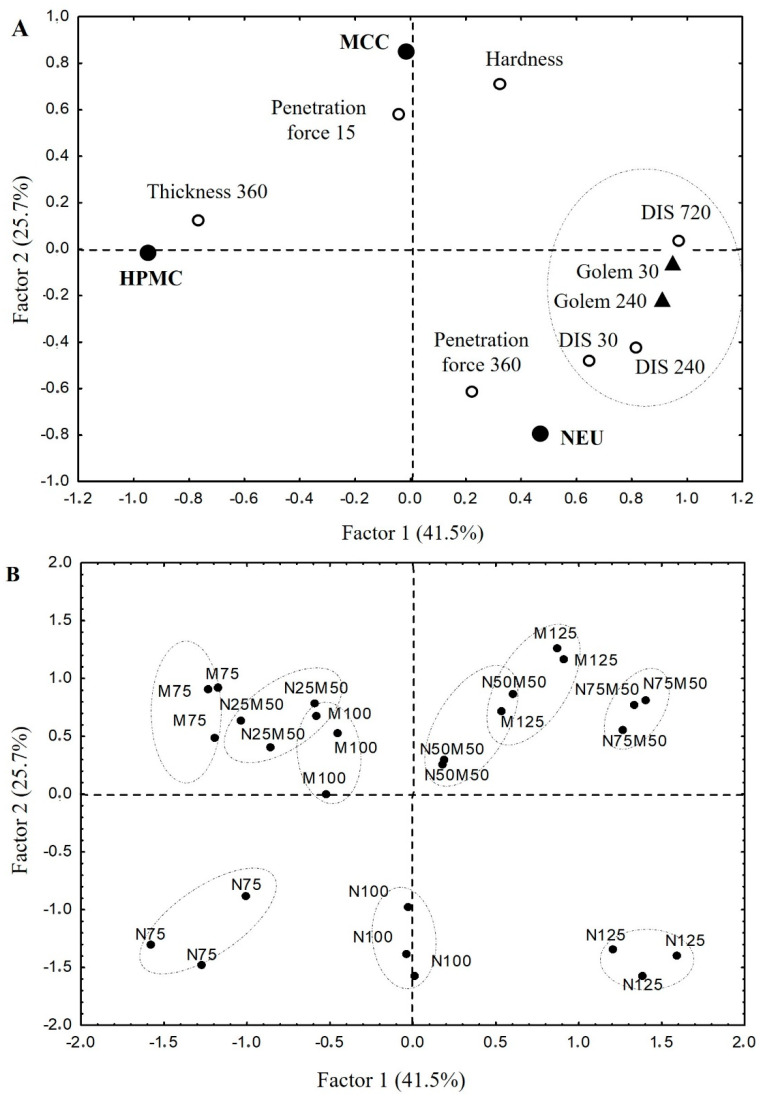
PCA loadings and scores plot: (**A**) PCA loadings plot; model depicts following variables: tablet hardness; drug amount released in 30 min = *burst effect*; drug amount released in 240 and 720 min; the gel layer thickness after 360 min swelling; penetration force in 15 and 360 min; (**B**) PCA scores plot; model shows a layout of the evaluated samples.

**Figure 9 pharmaceutics-14-00127-f009:**
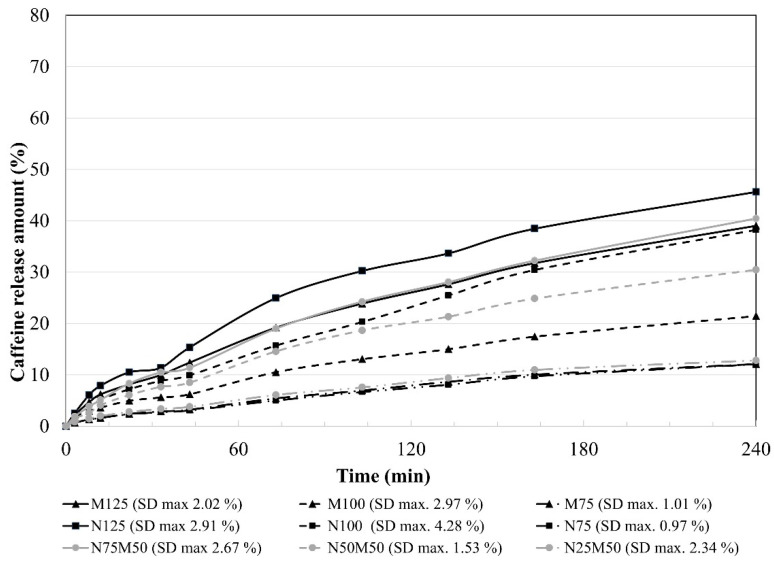
Cumulative dissolution data from all four phases of the dynamic dissolution test.

**Table 1 pharmaceutics-14-00127-t001:** Matrix tablets composition *.

Sample **	Caffeine		HPMC K4M		MCC PH 102		Neusilin^®^ US2	
	(mg)	(%)	(mg)	(%)	(mg)	(%)	(mg)	(%)
M125	100.0	38.8	25.0	9.7	125.0	48.5	0.0	0.0
M100	100.0	38.8	50.0	19.4	100.0	38.8	0.0	0.0
M75	100.0	38.8	75.0	29.1	75.0	29.1	0.0	0.0
N125	100.0	38.8	25.0	9.7	0.0	0.0	125.0	48.5
N100	100.0	38.8	50.0	19.4	0.0	0.0	100.0	38.8
N75	100.0	38.8	75.0	29.1	0.0	0.0	75.0	29.1
N75M50	100.0	38.8	25.0	9.7	50.0	19.4	75.0	29.1
N50M50	100.0	38.8	50.0	19.4	50.0	19.4	50.0	19.4
N25M50	100.0	38.8	75.0	29.1	50.0	19.4	25.0	9.7

* Each sample contains 0.5% of Aerosil^®^ 200 and 2.5% of magnesium stearate for better flowability and compression feasibility. ** The HPMC amount always corresponds to 150 minus the numeric value or the sum of the numeric values in the sample designation.

**Table 2 pharmaceutics-14-00127-t002:** The main characteristics of the powder blends and the matrix tablets.

Sample	Hausner Ratio ± SD	Average Weight ± SD [mg]	Hardness SD ± [N]	Friability [%]	Average Content ± SD [%]
M125	1.33 ± 0.04	256.7 ± 0.0010	129.8 ± 3.80	0.30	103.13 ± 1.75
M100	1.37 ± 0.01	251.9 ± 0.0038	107.8 ± 3.90	0.28	104.65 ± 3.08
M75	1.39 ± 0.00	256.5 ± 0.0011	133.0 ± 4.00	0.14	108.49 ± 3.67
N125	1.28 ± 0.00	261.7 ± 0.0035	118.2 ± 3.30	0.18	103.01± 2.57
N100	1.30 ± 0.00	250.3 ± 0.0031	98.2 ± 5.90	0.19	104.93 ± 1.48
N75	1.30 ± 0.01	250.1 ± 0.0020	105.1 ± 6.70	0.13	103.78 ± 1.24
N75M50	1.29 ± 0.01	255.6 ± 0.0023	156.0 ± 5.90	0.23	103.77 ± 3.99
N50M50	1.32 ± 0.01	251.3 ± 0.0019	142.5 ± 6.40	0.21	108.07 ± 2.71
N25M50	1.37 ± 0.01	257.4 ± 0.0021	139.2 ± 5.90	0.23	107.54 ± 4.61

**Table 3 pharmaceutics-14-00127-t003:** Mathematical modelling and drug release kinetics from matrices.

	Determination Coefficients for Mathematical Models							
	Higuchi	Korsmeyer–		Zero-	First-	Weibull		Included Time Interval (min)
		Peppas		Order	Order			
Sample	*R* ^2^	*R* ^2^	Release Exponent *n*	*R* ^2^	*R* ^2^	*R* ^2^	*b*	
M125	0.9572	0.9983	0.515	0.2929	0.9801	0.9979	0.73	540
M100	0.921	0.9928	0.485	0.1304	0.9865	0.9983	0.78	540
M75	0.9375	0.9924	0.508	0.26	0.9906	0.9975	0.82	540
N125	xx	0.99	0.307	xx	0.8842	0.9965	0.52	240
N100	0.8125	0.9924	0.561	xx	0.9902	0.9955	0.89	240
N75	0.8821	0.9899	0.573	0.1626	0.9948	0.9944	1.01	240
N75M50	0.9916	0.9991	0.568	0.6066	0.9861	0.9997	0.73	720
N50M50	0.9963	0.9978	0.543	0.6602	0.9832	0.9999	0.66	720
N25M50	0.9821	0.9915	0.494	0.5366	0.9885	0.999	0.77	720

**Table 4 pharmaceutics-14-00127-t004:** Similarity factor analysis.

Dissolution Apparatus.	USP 2		Biorelevant		USP 2 vs. Biorelevant		
Compared Samples	*f* _2_		*f* _2_		Compared Samples	*f* _2_	
M125/N75M50	73.05	S *	95.29	S	M125/M125	47.48	N
M125/N125	37.94	N **	66.94	S	N75M50/N75M50	58.66	S
N75M50/N125	34.35	N	67.81	S	N125/N125	29.28	N
M100/N50M50	69.52	S	66.24	S	M100/M100	40.01	N
M100/N100	45.96	N	55.19	S	N50M50/N50M50	55.45	S
N50M50/N100	47.2	N	73.14	S	N100/N100	40.31	N
M75/N25M50	83.63	S	94.54	S	M75/M75	36.31	N
M75/N75	68.52	S	99.29	S	N25M50/N25M50	37.9	N
N25M50/N75	61.95	S	96.87	S	N75/N75	38.98	N

* similar dissolution profile, ** non-similar dissolution profile.

**Table 5 pharmaceutics-14-00127-t005:** Penetration force of gel layer of swollen matrices measured by texture analyser for 6 h.

Sample	Penetration Force (N)								
Time Point (min)	15	30	60	120	180	240	300	360	Max. SD
M125	0.72	0.71	0.69	0.60	0.71	0.64	0.61	0.65	0.13
M100	0.69	0.68	0.60	0.61	0.58	0.60	0.57	0.59	0.09
M75	0.76	0.60	0.58	0.57	0.55	0.61	0.71	0.56	0.09
N125	0.63	0.62	0.67	0.68	0.70	0.74	0.72	0.72	0.04
N100	0.63	0.59	0.61	0.71	0.72	0.71	0.73	0.66	0.06
N75	0.69	0.65	0.58	0.61	0.63	0.65	0.71	0.73	0.06
N75M50	0.76	0.63	0.64	0.66	0.68	0.69	0.64	0.63	0.05
N50M50	0.68	0.60	0.58	0.58	0.57	0.55	0.57	0.55	0.08
N25M50	0.68	0.59	0.58	0.64	0.58	0.55	0.52	0.51	0.04

**Table 6 pharmaceutics-14-00127-t006:** Mathematical modelling and drug release kinetics from matrices (biorelevant dynamic dissolution).

	Determination Coefficients for Mathematical Models							
	Higuchi	Korsmeyer—Peppas		Zero-Order	First-Order	Weibull		Included Time Interval (min)
Sample	*R* ^2^	*R* ^2^	Release Exponent *n*	*R* ^2^	*R* ^2^	*R* ^2^	*b*	
M125	0.9655	0.9966	0.64 (0.61; 0.68)	0.8862	0.9889	0.9973	0.73 (0.62; 0.85)	240
M100	0.9669	0.9912	0.63 (0.57; 0.68)	0.8654	0.9722	0.9901	0.63	240
M75	0.9426	0.9931	0.70 (0.64; 0.76)	0.9243	0.9906	0.9935	0.81 (0.62; 0.99)	240
N125	0.9685	0.9884	0.61 (0.55; 0.68)	0.8450	0.9873	0.9920	0.80 (0.62; 0.97)	240
N100	0.9336	0.9962	0.73 (0.69; 0.78)	0.9463	0.9891	0.9957	0.73	240
N75	0.9356	0.9949	0.72 (0.67; 0.78)	0.9393	0.9885	0.9949	0.72	240
N75M50	0.9530	0.9967	0.68 (0.64; 0.72)	0.9145	0.9931	0.9969	0.79 (0.69; 0.90)	240
N50M50	0.9499	0.9943	0.68 (0.63; 0.73)	0.9135	0.9942	0.9967	0.84 (0.71; 0.97)	240
N25M50	0.9569	0.9925	0.66 (0.60; 0.72)	0.8948	0.9875	0.9930	0.76 (0.58; 0.95)	240

## Data Availability

The datasets corresponding to the current study are available from the corresponding author upon request.

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
