# Peer review of "Exploration of Neusilin® US2 as an Acceptable Filler in HPMC Matrix Systems—Comparison of Pharmacopoeial and Dynamic Biorelevant Dissolution Study"

_pharmaceutics, 2022, doi:10.3390/pharmaceutics14010127_

Round 1
Reviewer 1 Report
The reviewer agreed to recommend this article for publication, but there are some small problems to be noted:
- The author needs to be rigorous about the format of the article, such as line 42
- In Figure 4, sufficient distance is required between each image to distinguish different samples
- Figure 5, there is only one sample per point in this figure? Were parallel samples needed to count SD values?
- Figure 7, The distance between each image to looks casual.
- Figure 6, also need SD value?
Author Response
Response to Reviewer 1 Comments
IN red:
We greatly appreciate the positive evaluation of our manuscript. Thank you a lot for your time and all your comments and suggestions. You can see below our answers. The changes in the manuscript body are in red.
The reviewer agreed to recommend this article for publication, but there are some small problems to be noted:
- The author needs to be rigorous about the format of the article, such as line 42
We adjusted the format of the whole manuscript.
- In Figure 4, sufficient distance is required between each image to distinguish different samples
We changed the distance between the pictures (Line 266).
- Figure 5, there is only one sample per point in this figure? Were parallel samples needed to count SD values?
Six units were evaluated for each sample. Mean average and SD were calculated for each time point from these six units. We state only the SDmax for each sample, as the SDs were low, and they would also disrupt the clarity of the graph, as multiple profiles are in one image. However, we have tweaked the text about the six units per sample (Methods section) for better clarity (Lines 144-146)
- Figure 7, The distance between each image to looks casual.
We adjusted the distance between the figures – Line 389.
- Figure 6, also need SD value?
We added the maximum SD values for all tested samples behind the sample's name. We decided this way due to low clarity if SD intervals are part of the graph. We have also adjusted the methods section (Lines 214-215) to clearly distinguish between the sample and the measured units.
Reviewer 2 Report
The manuscript deals with the evaluation of the use of the excipient Neusilin US2 as a filler in HPMC matrices in terms of dissolution studies using USP apparatus II and Golem v2 apparatus. The manuscript is of interested and experimental work well constructed. Some few points are:
What is “a commonly used indifferent filler” (Abstract 3rd line)?
For what application the use of Neusilin US 2 as a filler is intended for? It is not clear, according to which Neusilin can be considered a suitable filler or not?
How formulations in Table 1 were selected?
Differences and implications of using the two dissolution apparatus should be better highlighted.
Author Response
Response to Reviewer 2 Comments
In green:
We greatly appreciate the positive evaluation of our manuscript. Thank you a lot for your time and all your comments and suggestions. You can see below our answers. The changes in the manuscript body are in green.
The manuscript deals with the evaluation of the use of the excipient Neusilin US2 as a filler in HPMC matrices in terms of dissolution studies using USP apparatus II and Golem v2 apparatus. The manuscript is of interested and experimental work well constructed. Some few points are:
- What is "a commonly used indifferent filler" (Abstract 3rd line)?
We apologize for this unsuitable term. We meant "inert "
- For what application the use of Neusilin US 2 as a filler is intended for? It is not clear, according to which Neusilin can be considered a suitable filler or not?
We highlighted the results of our experimental work in the abstract and Conclusion parts (in blue color)
- How formulations in Table 1 were selected?
The selection of formulations followed our experimental work (DOI: 10.1155/2019/8043415 - https://www.hindawi.com/journals/bmri/2019/8043415/, and DOI: 10.1208/s12249-017-0870-6 https://pubmed.ncbi.nlm.nih.gov/28971441/) in which we used Neusilin for application of water dispersion of insoluble Eudragit polymers. We noticed the excellent properties of Neusilin US2 during experimental work and decided to explore their own influences on HPMC systems in detail. Moreover, we are also investigating the solidification of self-emulsifying systems, and these results could be a valuable tool also in this field.
- Differences and implications of using the two dissolution apparatus should be better highlighted.
In results and discussion section 3.4, more precise and explaining information was added in Lines 487-490 to improve this shortcoming. Also, the conclusion output for this problematics was enhanced in the wholly rewritten Conclusion.
Reviewer 3 Report
The authors present a study of Neusilin US2 as an alternative to MCC as a matrix filler. The study is accompanied by an extensive dissolution rate evaluation by USP and biorelevant media methods. The work is interesting to formulation scientists. However, what seems to be missing from this work, is a quantitative evaluation of the relative contributions of matrix erosion and swelling (diffusion layer) in the dissolution mechanism. This is a point that requires consideration, and since the addition of matrix erosion studies is rather laborious task, at least the discussion of the possible role of erosion on drug release, should be more extensive.
Other than that, the authors should stress out the novelty of their work, since the main findings aren't unexpected, considering the behavior of each excipient upon hydration: High concentrations of Neusilin probably reduce the HPMC matrix's capacity to form a continuous gel and induce erosion, while in lower concentrations, and combined with MCC, the integrity of the gelling matrix is preserved, wile the gel layer is favorably modulated to provide zero order release kinetics.
Some minor points are listed below:
Abstract, line 3: what is an "indifferent filler"? Do the authors mean "inert"?
Abstract, references to a commercial apparatus (GOLEM) should be restricted to the Methods section and not here.
Figure 9, M125 at 35 minutes the dissolution presents a plateau and then an exponential rise. Unless this is due to experimental variation, how is it explained and why isn't such a behavior observed for other formulations?
Author Response
Response to Reviewer 3 Comments
In blue.
We greatly appreciate the positive evaluation of our manuscript. Thank you a lot for your time and all your comments and suggestions. You can see below our answers. The changes in the manuscript body are in blue.
The authors present a study of Neusilin US2 as an alternative to MCC as a matrix filler. The study is accompanied by an extensive dissolution rate evaluation by USP and biorelevant media methods. The work is interesting to formulation scientists. However, what seems to be missing from this work, is a quantitative evaluation of the relative contributions of matrix erosion and swelling (diffusion layer) in the dissolution mechanism. This is a point that requires consideration, and since the addition of matrix erosion studies is rather laborious task, at least the discussion of the possible role of erosion on drug release, should be more extensive.
A more extensive discussion of erosion role was added – Line 334-344, 514-522.
Other than that, the authors should stress out the novelty of their work, since the main findings aren't unexpected, considering the behavior of each excipient upon hydration: High concentrations of Neusilin probably reduce the HPMC matrix's capacity to form a continuous gel and induce erosion, while in lower concentrations, and combined with MCC, the integrity of the gelling matrix is preserved, while the gel layer is favorably modulated to provide zero-order release kinetics.
We stress out the novelty of our work in the abstract and Conclusion part.
Some minor points are listed below:
- Abstract, line 3: what is an "indifferent filler"? Do the authors mean "inert"?
We apologize for this unsuitable term. We meant "inert" (in green)
- Abstract, references to a commercial apparatus (GOLEM) should be restricted to the Methods section and not here.
Thank you very much for this comment. We removed it from the text of the abstract. Line 24, 30. Moreover, we restricted references to a commercial apparatus to the Methods except for Figure 8 and one mention in Section 3.4. Biorelevant dissolution study (Line 481)
- Figure 9, M125 at 35 minutes the dissolution presents a plateau and then an exponential rise. Unless this is due to experimental variation, how is it explained and why isn't such a behavior observed for other formulations?
The reviewer probably meant N125 - We added a paragraph to the manuscript's text explaining this observation – Line 508-513.
Reviewer 4 Report
The authors investigated the impact of NEU in HPMC matrix system w/wo MCC. Interesting results were found for the NCUM/MCC combination samples. The paper is on a topic of importance and will be of interest to others working in the field. I recommend publication with minor changes.
- The authors should provide detailed information of the powder properties, such as density and particle size distribution.
- Hausner ratio is a rough indicator to determine powder flow properties. I suggest that the authors use FT4 rheometer and/or Schulze shear test to measure the powder flow properties
- The authors should provide detailed tableting conditions. I suggest that the authors add one figure to show the compressibility profile (tablet density vs. compression forces), and compactibility profile (tablet hardness vs. compression forces). It would be interesting to see the impact of different tableting conditions on the dissolution profile.
Author Response
Response to Reviewer 4 Comments
We thank the reviewer for the comments. The powders were blended from well-known excipients at defined conditions we deem as easily replicable. The suggested flow properties devices are not at our disposal, and though the Hausner ratio is a bit cruder indicator, it still belongs to the pharmacopeial testing and it is possible to draw na outline. The last part is a great suggestion, however this would require completely new preparations and measurements, which is impossible to catch up in time. In the present study we compressed to the maximum force possible. However, in our further study, as suggested in conclusion, we would like to adopt also this suggested approach a investigate different tableting conditions.